# Urban Traffic Noise Analysis Using UAV-Based Array of Microphones

**DOI:** 10.3390/s23041912

**Published:** 2023-02-08

**Authors:** Marius Minea, Cătălin Marian Dumitrescu

**Affiliations:** Department Telematics and Electronics for Transports, University Politehnica of Bucharest, 060042 Bucharest, Romania

**Keywords:** sound pressure level, noise pollution, UAV data collection, intelligent measurement of traffic noise

## Abstract

(1) Background: Transition to smart cities involves many actions in different fields of activity, such as economy, environment, energy, government, education, living and health, safety and security, and mobility. Environment and mobility are very important in terms of ensuring a good living in urban areas. Considering such arguments, this paper proposes monitoring and mapping of a 3D traffic-generated urban noise emissions using a simple, UAV-based, and low-cost solution. (2) Methods: The collection of relevant sound recordings is performed via a UAV-borne set of microphones, designed in a specific array configuration. Post-measurement data processing is performed to filter unwanted sound and vibrations produced by the UAV rotors. Collected noise information is location- and altitude-labeled to ensure a relevant 3D profile of data. (3) Results: Field measurements of sound levels in different directions and altitudes are presented in the paperwork. (4) Conclusions: The solution of employing UAV for environmental noise mapping results in being minimally invasive, low-cost, and effective in terms of rapidly producing environmental noise pollution maps for reports and future improvements in road infrastructure.

## 1. Introduction

Urban traffic is, in present days, a major factor of urban environmental pollution. Therefore, most of the EU countries and countries worldwide are trying to alleviate the negative effects of traffic on the environment, seeking solutions both in infrastructure and on the mobile platforms. Both emissions of traffic-generated chemical composites, particulate matter, and noise produce negative effects on residents, traffic participants, and local natural life. Knowing the way that noise from daily road traffic is dispersed and behaves could be beneficial for competent authorities and municipalities to evaluate and propose countermeasures to reduce the impact on environment and citizens’ lives. In this paper, research that has been carried out is presented to determine the main characteristics of the urban noise produced by traffic, and a solution for a cost-effective and efficient collection of noise data is also presented. The data collection is carried out using a UAV-airborne set of microphones and a data processing unit.

The research in this field is quite rich, and several solutions for collecting information on environmental noise pollution have been proposed by different scholars and industrial researchers.

Starting with modeling, both polluting emissions and noise assessment in urban areas represent a measure designed to find the most appropriate solutions for reducing their impact on citizens’ lives, as shown in [1], where Hui Di et al. determined in specific conditions a traffic noise propagation model to predict instantaneous sound levels, based on noise attenuation. The authors consider that noise is affected by air absorption, obstacle reflex, and ground absorption during its transmission process, which causes noise attenuation, and these elements are considered in building a noise mapping and assessment model. In their conclusion, the authors show that the quality of the acoustic environment seems to be more moderate during the daytime and better at night.

Zefreh, M. M. and Torok, A. go deeper in the analysis of noise and consider different traffic conditions to evaluate their effect on noise intensity and dispersion [2], performing theoretically generated conditions by the Monte Carlo Simulation technique, following the distribution of traffic speed in the urban roads. They show in the conclusions that “the free flow condition and the oversaturated condition would generate the same amount of sound energy (sound exposure level) but the vehicles in free flow condition produce higher range of sound pressure level (58.7 dBA to 64.59 dBA)”.

Kai Cussen et al. [3] evaluated the noise produced by a UAV according to the legally mandated ISO 3744 and investigated the suitability of commercial implementations of ISO 9613 for modeling noise emission from UAVs. Finally, they state that models are adequate for assessing UAV’s noise in terms of directivity. In addition, they showed that most UAVs in the actual industrial production may exceed EU limits by around 1.8 dBA, causing possible urban noise issues.

The study published by Fiedler and Zannin [4] was performed in several Latin America cities to determine noise presence and produce noise maps, also including noise-sensitive areas and locations, such as hospitals. The study also shows that noise maps are extremely helpful tools in the processes of diagnosis and evaluation of noise pollution. The researchers employed different scenarios in hypothetical situations to produce and assess the effects of noise pollution on the environment, simulating increase and decrease of traffic volumes by 50–56% and studying the effects on noise production. At the end of their work, they concluded that reducing heavy traffic, or general traffic, by around 50% can produce a correspondent noise reduction of around 3 dBA (or dB(A)—A-weighted decibel that represents an expression of the relative loudness of sounds as perceived by the human ear; the latest standard is ISO 226:2003).

Doygun H. and Gurun, D.K. [5] tried to quantify noise pollution generated by the urban traffic in a Turkish city, producing over 114 measurements of noise levels in 38 different urban locations, classified as residential or industrial areas. The study was aimed at quantifying the temporal and spatial dynamics of urban traffic noise, to compare levels with national and international limits, and determine mitigative measures against noise pollution. Remarkably, in this study is the conclusion that noise pollution is usually ignored in public policies, regulations, and laws regarding environmental conditions. The authors recommend renewing road surfaces to reduce noise levels by around 5 dBA. Similar research is provided in papers [6,7,8,9]. High capacity and speed roads also came into focus for environmental emissions, PMs, and noise evaluation. In [10] is presented a study to gauge the existing public’s attitude and degree of awareness to contemporary vehicular noise pollution. The study revealed that most people are affected by traffic noise, several people attributing increased headaches and stress to the excessive noise levels. It has been shown by field evaluation that “air pollution and smoke disturb people with the percentage of 54.2% and 62.9%. From Pie chart 3 noise pollution also disturbs people surrounding with 60%”. In conclusion, the study recommends reducing traffic noise on urban highways by a program of shared responsibility. Similar information is presented in works [11,12].

Ciaburro G. et al. [13] expanded noise data collection purposes beyond simple environmental effect, giving it even more importance: early detection of security problems related to citizens’ mobility, crime, risk of terrorism. Their study has led to the conclusion that it is important to analyze the sound from different characteristics and tonal components. The ambient noise in UAV and non-UAV scenarios is very complex, it is not possible to distinguish between the different acoustic sources, at least in the time domain, and it is possible to employ a CNN-based classification system to identify the presence of UAVs with an accuracy of 0.91. Research is conducted in several papers to study effects of road gradients’ influence on chemical pollutant emissions (via modeling) and noise. Studies in similar areas of research are shown in papers [14,15,16,17].

The research conducted by Luo L. et al. [18] is oriented, on the other hand, towards collecting noise information via wireless acoustic sensor networks (WASNs). They propose a new system that employs WASNs to monitor the urban noise and recognize acoustic events with a high performance; the system is composed of sensors with the ability to produce local signal processing, convolutional neural networks for classification, and a platform for noise maps visualization. This direction of research is also employed by other authors, as shown in papers [19,20,21,22,23,24,25], employing also modeling of emissions based on Google maps road profiles [24].

Based on noise measurements, evaluations, and mapping, noise reduction strategies can be imagined for different urban or extra-urban scenarios. Hsiao Mun Lee et al. [26] extended the study on traffic-generated noise to affected population, dividing it into group noise indicators (highly annoyed and sleep-disturbed people) and studying the effect of noise reduction measures on these segments of the population. The results of this study showed that installing noise absorbents and barriers is highly recommended because it significantly reduces the influence of noise on nearby located environment. However, this solution solves neither the problem of chemical pollutants emissions nor the PM, also generated by traveling vehicles [27]. Of course, there are different effects on those segments of the population, according to the measures taken. Similar research is given in works [28,29,30].

General model-based noise mapping in urban areas is the focus in paper [31], where Ye Liu et al. discuss the working principle of general model-based noise mapping using mobile crowdsensing and acoustic sensor networks (ASNs). The authors propose an interesting solution, that is, collecting sound information via smart devices carried by citizens in an urban area (smart watches, smart bracelets, etc.) and using GPS-related information to send relevant data regarding levels and features of noise pollution. However, this solution would probably need serious discussions about ensuring privacy, as personal mobile devices are used to collect audio files. Connected with this research and with a view on future possible applications, based on work [32], noise-relevant data collection and, in a certain amount, chemical pollutants information could also be possibly collected by the means of connected vehicles. However, an overall view of existing technologies for noise data collection specific for smart cities is given in [33].

Within the collection of sound information in cities, it is also possible to detect specific events from this collection of data. Such a proposal is shown in work [34], where Soccoro J.C. et al. are investigating the possibility to eliminate anomalous noise events (ANE) unrelated to road traffic.

The preoccupation towards the employment of mobile crowdsourcing for data collection is also growing in this domain, as the new developments for smart city applications have issued mobile crowdsensing (MCS), a service designed to enable users to collect and share sensor data in different urban areas. As shown in [35], “MCS services can produce detailed sensor readings and provide means to discover new phenomena in urban environments that otherwise cannot be measured by individuals, such as the occurrence of traffic congestion, or environmental noise pollution monitoring”.

As it can be observed from the above literature review, the focus on collecting information regarding urban noise pollution is at high preoccupation of different researchers, and it is also in focus for the development and concept of new road infrastructures, especially in smart cities transformation processes. The purpose of the present research is to investigate a simple, but efficient and effective method for noise data collection employing a UAV that can be used both for building 3D noise maps and to detect ANEs on specific paths, contributing to the correct management of environmental protection and also to public security in large cities. As a future development, the solution of employing UAV-borne sensors could be also used for collecting relevant data on other types of road traffic-generated pollutants, such as CO, CO_2_, NOx, SO_2_, and PM, most of the latter deriving from harmful products, such as harmful byproducts including nitrogen dioxide, carbon monoxide, hydrocarbons, benzene, and formaldehyde.

The remainder of this paper is organized as follows: Section 2 is dedicated to Materials and Methods; Section 3, Results; Section 4, Discussion; and, finally, Conclusions.

## 2. Materials and Methods

### 2.1. General Information

Sound data collection is important in many applications for smart cities: control of environment, reduction of traffic-generated pollution, or even detection of security-related public events. Mapping the sound levels and characteristics could also be beneficial for smart city administrators, helping them to better control traffic flowing, delicate environmental areas such as hospitals, and people’s mobility, and to take the most appropriate measures to ensure a healthy level of ambient noise in urban areas to provide the best environment for living.

Sound pressure level (SPL) represents the difference in the local ambient atmospheric pressure that is caused by the sound wave. Having as a measurement unit the Pascal (Pa), the SPL can be measured in open air using a microphone. Usually, a microphone’s sensitivity is estimated using a sine wave at 94 dB SPL (equivalent to 1 pascal of air pressure) at the frequency of 1 kHz, where the human acoustic sensitivity is high. The microphone’s sensitivity is measured as its response to the 94 dB SPL signal. Such a microphone considers the maximum value of sound pressure (measured in dB SPL) at which the microphone can work correctly before producing audible levels of distortion. Close-passing road traffic usually produces noise ranging from 60 to 100 dB SPL (dB SPL=20log10P1P0−P1, where $P_{1}$ represents the sound pressure level measured [Pa] and $P_{0}$ is the reference sound pressure level of 20 [µPa]). The value of 0 dB SPL corresponds to the lowest threshold of human hearing. In normal conditions, pressure levels might reach quite a large range, spreading from 0.00002 to 200 pascals. However, sensors and devices such as microphones do not have the same dependency on SPL as the human ears. Therefore, A-weighting represents a way to make the sound loudness relative to SPL to be perceived by sensors in a similar way as the human ears. While the theoretical sound frequency band perceived by the human ears is 20–20,000 Hz, in reality, it depends quite a lot on individuals, and especially on their age. Older people usually have a much less broad band perception. However, in the 20–20 kHz frequency range, a normal person perceives lower frequencies with lower sensitivity, and human hearing is generally logarithmic in nature, so decibels, as a measurement unit, match in a large amount with human perception and experience. Usually, normal people perceive sounds with a maximum accuracy in the range of 250–5000 Hz. Therefore, a so-called “A-weighted” curve is used to compensate an acoustic sensor’s sensitivity and make it similar, or at least very close to that of the human ears. The A-weighted curve is depicted in Figure 1a, and the conversion from [dB SPL] to [dBA] employing A-weighting adjustments is presented in Figure 1b.

A-weighting is the standard for establishing hearing damage caused by high sound pressure levels and also used to determine noise pollution. International Electrotechnical Commission standard 61672:2003 presents the curves that are employed in sound meters. All modern sound meters are able to provide A-weighted decibel and C-weighted decibel (dBC) measurements (C-weighted is a flatter curve of sensitivity, used beyond 100 dBs, where the human hearing characteristics also change). In this standardization, there is also the “Z-weighted” curve, a specific characteristic where no compensation is applied, defining the band between 20 Hz and 20 kHz with ±1.5 dB.

SPL, in decibels, is described by Equation (1), as defined by ISO 9613-2:1996 standard.
(1)$$L_{AT}=10\mathrm{lg}\left\{\left[\left(\gamma T\right)\int_{0}^{T}p_{A}^{2}\left(t\right)dt\right]/p_{0}^{2}\right\}$$
where:

$L_{AT}$—equivalent continuous A-weighted sound pressure level;

$p_{A}\left(t\right)$—the instantaneous A-weighted sound pressure, [Pa];

$p_{0}$—the reference sound pressure (=$20\times 10^{-6}$) Pa;

T—specified time interval [s].

The attenuation produced by geometrical sound divergence may be computed according to Equation (2):(2)$$A_{Div}=\left[20lg\left(\frac{d}{d_{0}}\right)+11\right]$$
where:

*d*—distance from source to receiver [m];

$d_{0}$—reference distance (1 m).

The atmospheric absorption is given by:(3)$$A_{atm}=\frac{\alpha d}{1000}$$
where *α*—the atmospheric attenuation coefficient [dB/km].

### 2.2. Instruments and Initial Setup

In order to initially determine the main characteristics of the traffic-generated noise, some field measurements have been conducted. The instrumentation employed for this purpose was composed of:
-hardware: sonometer (UNI-T UT 353BT) with a Bluetooth connection to a smartphone for data storage and processing) and a smartphone Android (Samsung Galaxy A50 SM-A505FN/DS);-software: UTienv 2.0.8 and Sound Analyzer 2.4 running on Android 11.


The tests were performed in a busy junction in Bucharest, Romania (Figure 2).

The location for initial measurements was chosen considering that the respective area is known as very busy and with high levels of traffic-generated noise. The main boulevard is four lanes per direction of traveling, and there are also public transport lines. The location is situated near a very busy subway station and a large market. Ground level initial measurements of traffic-generated noise had the purpose of determining actual levels of noise pollution, frequency range, and main causes. The microphone was placed 1 m above ground, on a tripod, facing traffic, 5 m from the first lane (on a shoulder of the road). Post-processing of collected information consisted of recording levels on different frequencies and analyzing SPL.

However, due to the current period, there were some specific weather conditions when the tests have been performed, meaning the carriageway was wet, causing increased traffic noise with a slightly different frequency range, compared to noise generated by the normal (dry) roadway surface. Still, we consider that this is not a major downside, as, in this case, we managed to collect information in a really peak noise environment, in a very busy area. Further field measurements will be taken also in different humidity conditions, to compare results. Outside temperature was 3 °C. The area is surrounded in the N–E and S by relatively high buildings, favoring propagation of noise with many reflections. The NW part is occupied by a busy market and commercial areas.

### 2.3. Initial Findings

Based on the field-collected information, several diagrams have been drawn to determine the main components of traffic-generated noise and their amplitudes, knowing the fact that the main health affections caused by traffic noise are increased hypertension risk, myocardial infarction, cardiovascular, and even stroke mortality risk. High noise sensitivity has also been recently associated with lower ischemic heart disease mortality risk, and road traffic-generated noise was also associated with Phase 4 psychological ill-health, but only among those exposed to 56–60 dBA [36]. In the area where the initial measurements were conducted, there are a lot of buildings with numerous flats, so it also represents an important residential area, hence the importance of finding such noise levels. Initial results are presented in the following diagrams.

It can be observed in Figure 3 that the main traffic noise components (with SPL greater than 50 dBA) increase from 100 Hz and extend up to over 15 kHz, occupying most parts of the human hearing spectrum. The peak levels are reached at around 400–500 Hz. An interesting fact is that some medical reports and scientific papers claim that a 432-Hz tuning has better effects on the human body, but there are no scientific studies that support this hypothesis. The sound frequency of “432 Hz … was associated with a slight decrease of mean (systolic and diastolic) blood pressure values (although not significant), a marked decrease in the mean of heart rate (−4.79 bpm, *p* = 0.05) and a slight decrease of the mean respiratory rate values (1 r.a., *p* = 0.06), compared to 440 Hz” [37], but this is the case for music, and not for incoherent sounds. Usually, traffic noise is responsible for stress increase.

To determine more precisely the sound components, separate diagrams have been produced for the main harmonic frequencies, as shown in Figure 4, Figure 5 and Figure 6.

When analyzing the diagrams in Figure 4, Figure 5 and Figure 6, another observation that may be drawn is that the levels are most varied in the lower frequency range than in medium and higher frequency ranges (Figure 4 compared to Figure 5 and Figure 6).

Levels appear to be more constant in midrange frequencies (Figure 5), exceeding 55 dBA on a regular basis. Still, considering that the initial field measurements have been conducted with a wet roadway surface, it is possible that a higher range of frequencies (above 2 kHz) of the levels of sound pressure are to be favorized by the humid surface of roads. Finally, sound components recorded for higher frequencies also show an increased variation compared to medium range (Figure 6 compared to Figure 5).

To determine changes in sound characteristics when picking up data at a different altitude, another set of field measurements has been conducted using a flat in a nearby building at 30-m high by placing the instrumentation in this location. The results are presented in the following diagrams.

In Figure 7, recorded higher frequencies experienced a significant decrease, up to 40 dBA, while the peak changed also in frequency, being situated around 1900 Hz. In addition, while the peak level of sound pressure remained around 60 dBA, this was only for a smaller band, from approx. 1 kHz up to 5 kHz, and the rest of the levels experienced a decrease, more obviously in the lower domain.

One observation that is worth mentioning is that noise levels are more variable when recorded at higher altitude; this is probably because of side activities’ influence (especially in the 50-Hz range, Figure 8).

This remarque is also available for the other frequencies (medium range and higher range, Figure 9 and Figure 10).

From the above presented measurements, it results that it is possible to collect relevant information on traffic-generated noise even from above, meaning that a solution to provide a more detailed imaging on sonic pollution is possible to obtain from an automated UAV flight over selected areas, at specific intervals of time.

### 2.4. Information Collection Process

Data collection regarding traffic-generated noise is important in the development of a friendly environment for the inhabitants of a large city. However, it is not economic to install sensors in every important point of a street’s network, and there are involved important costs in such as an investment. Still, it is advisable that the next generation for this activity employ sensors in a future connected vehicles network, installed onboard vehicles, using those as “floating cars”, or moving sensors, in the traffic flow. Until then, though, a simpler solution is proposed in this research—to employ periodic measurements in different locations using a UAV. The vehicle is equipped with a specific configuration of microphones, able to create a very informative sound imaging and overall information regarding the sonic pollution that moving traffic is producing in large urban agglomerations.

The proposed solution estimates traffic noise based on the use of a drone on which a directional microphone array is mounted. To reduce the background noise, and especially the noise produced by the propeller rotations, an adaptive noise cancellation (ANC) filter developed on the FPGA is employed. Two microphones are pointed upwards, collecting noise from UAV rotors, in order to reduce its level. Adaptive filtering is performed with an optimized Least-Mean-Square (LMS) algorithm. The general architecture of the system is presented in Figure 11, and, in the following subsections, the mathematical concepts and processing modules are described.

### 2.5. Sound Processing

The directional microphone array is arranged on a Kintex 7 FPGA development board with a square-shaped ultrasound sensor array containing 24 MEMS capsules, spaced at 11 mm. Figure 12 shows the simulation of the square microphone array together with the beamforming analysis using MUSIC DOA (Multiple Signal Classification & Direction of Arrival) [38]. DOA indicates the direction from which a propagating wave typically arrives at a point where a set of sensors is placed [39]. The image on the left of Figure 12 shows the energy detection of the acoustic signal generated by the drone’s rotors, along with the location position labeling (azimuth and elevation), for the two acoustic frequencies characteristic of drones (white color), represented on the frequency spectrum (bottom right).

The configuration of the acoustic processing system is presented in Figure 13—the acoustic signal picked up from the microphone sensor array is converted from pulse density modulation (PDM) to pulse code modulation (PCM) for each sensor. The conversion module consists of a suite of digital filters, and, finally, a multiplication block is inserted (as shown in Figure 14). PDM can be the ideal choice in challenging application environments, thanks to its inherent resilience to noise.

In the first stage, a low-pass comb filter for PDM to PCM conversion is employed. To achieve lossless conversion, we use a half-band compensation filter because the passband of the comb filters is not flat, and the last digital filter of the finite impulse response (FIR) type has the role of eliminating the noise remaining after conversion. Finally, the multiplier converts the signal to a frequency in the audible spectrum. The sampling frequency for PDM is 4.608 MHz for the MEMS sensors used in the construction of the microphone array.

### 2.6. Delay and Sum Beamforming

To achieve the Delay & Sum mode, we use the delay and sum algorithm (DAS)–Bartlett beam. The formation of the DAS beamformer consists of applying a delay and a weight to the output of each sensor, and, finally, summing the signals [40]. The delays have the role of obtaining the maximum of the sensitivity matrix for the received acoustic signals. Moreover, by adjusting the delays, the directionality of the microphone area is oriented towards the direction of interest for the reception of the signals, thus achieving the consecutive addition of the individual acoustic signals received from the sensors. This aspect means that, at certain angles of incidence, constructive interference is obtained, and, for other angles of incidence, destructive interference is obtained. Figure 15 shows the way the DAS algorithm works.

### 2.7. Directional Energy

To perform the audio energy analysis, some processing must be performed to identify the energy of the summed audio signals [39,40,41]. Because the energy of a continuous signal $x_{c}\left(t\right)$ is calculated with the formula:(4)$$E_{c}=\int_{-\infty }^{+\infty }{\left|x_{c}\left(t\right)\right|}^{2}dt$$
for discrete signals *x*(*n*), the energy is calculated with the formula:(5)$$E_{d}=\overset{+\infty }{\underset{-\infty }{\sum }}{\left|x\left[n\right]\right|}^{2}$$

The hardware implementation for the directional energy calculation is shown in Figure 16. An FPGA multiplier module is employed to form the square DAS beamforming signal. The output of the accumulator module calculates the signal energy based on the acquired samples.

### 2.8. Adaptive Filtering

The adaptive filter for identifying an unknown system has the configuration shown in Figure 17. In this configuration, *x*(*n*) represents the input signal at the discrete time at moment *n*, while *y*(*n*) is the output of the system to be determined (i.e., *h*) [41]. By filtering the input signal through both systems, we can write $\hat{y}\left(n\right)={\hat{h}}^{T}\left(n-1\right)x\left(n\right)$ and $y\left(n\right)=h^{T}x\left(n\right)$, where x(n) is a vector containing the most recent L input samples [42,43].

Finally, the output of the adaptive system is subtracted from the desired signal, so that the error signal *e*(*n*) results in $e\left(n\right)=d\left(n\right)-\hat{y}\left(n\right)$.

### 2.9. Least-Mean-Square (LMS) Algorithm

The Wiener filter facilitated the development of one of the most popular adaptive algorithms, the LMS algorithm. The popularity of the LMS algorithm is due to its ease of understanding, implementation, and use. At first, the Wiener filter aimed to determine an optimal filter, ${\hat{h}}_{w}$, which minimizes the error signal in each iteration [43,44].

In this context, the optimal filter that minimizes the error signal is identified by minimizing a cost function, the mean squared error (MSE), defined as $J\left[\hat{h}\left(n\right)\right]=E\left[e^{2}\left(n\right)\right]$, where E[.] represents the statistical averaging operator. After several calculations, the optimal Wiener filter is obtained as ${\hat{h}}_{W}=R_{x}^{-1}p_{xd}$.

Calculation of the $p_{xd}$ correlation vector and the autocorrelation matrix, $R_{x}$, involves the knowledge of some statistical estimates, making the steepest slope algorithm unsuitable for practical use [45,46]. If we consider the instantaneous values:(6)$$p_{xd}\left(n\right)=d\left(n\right)x\left(n\right),\;R_{x}\left(n\right)=x\left(n\right)x^{T}\left(n\right)$$
the following equation results:(7)$$\hat{h}\left(n\right)=\hat{h}\left(n-1\right)+2\mu x\left(n\right)e\left(n\right)$$
which defines the updated equation of the LMS algorithm.

The theoretical configuration of an adaptive filter for noise reduction/adaptive noise cancellation (ANC) is presented in Figure 18.

### 2.10. Optimized LMS Algorithm

To improve the performance of the LMS algorithm, the fixed step of updating the coefficients is replaced with a control matrix. *Y*(*n*) is considered the scaled unitary control matrix, and:(8)$$Y\left(n\right)=\mu \left(n\right)M_{L}$$
where $M_{L}$ is the *LxL* identity matrix and μ is fixed step size.

The autocorrelation matrix for the error coefficients is then calculated, replacing the misalignments, and obtaining the following result:(9)$$\begin{array}{c} R_{c}\left(n\right)=\left[R_{c}\left(n-1\right)+\sigma_{h}^{2}M_{L}\right]-\mu \left(n\right)\left[\left[R_{c}\left(n-1\right)+\sigma_{h}^{2}M_{L}\right]R_{x}+R_{x}\left[R_{c}\left(n-1\right)+\sigma_{h}^{2}M_{L}\right]\right]\\ +\mu^{2}\left(n\right)\left\{2R_{x}\left[R_{c}\left(n-1\right)+\sigma_{h}^{2}M_{L}\right]R_{x}+R_{x}tr\left[\left[R_{c}\left(n-1\right)+\sigma_{h}^{2}M_{L}\right]R_{x}\right]\right\}+\mu^{2}\left(n\right)\sigma_{w}^{2}R_{x} \end{array}$$
where:

$\sigma_{h}^{2}$—system noise variation;

$R_{x}$—autocorrelation matrix of the input signal.

The ANC simulation result with optimized LMS is presented in Figure 19. It is found that the system noise variation parameter has an important role because if it has a high value, it allows the optimized LMS algorithm to determine the new system parameters in a reasonable calculation time.

### 2.11. Neural Network for Road Traffic Noise Prediction

For the prediction of road traffic noise, a multilayer feed-forward neural network (MFFN) architecture is employed [46,47,48]. The classic structure of the MFFN is shown in Figure 20.

MFFNs are layered feed-forward networks, trained with statistical inverse propagation. The main advantages are determined by the ease of use and implementation, but also that they can approximate any input/output map [49,50]. The input information of the MFFN network for traffic volume, average speed, and vehicles is obtained from a convolutional neural network (CNN), and the gradient and building density information is determined by Equations (9) and (10) presented in the following methodology subsection.

## 3. Results

### 3.1. Methodology

The equalization of the continuous sound level, weighted A, is achieved by the constant sound level for a period of time and contains the same acoustic energy as its variation in real time [50,51,52,53,54]. Mathematical expressions are defined by:(10)$$L_{Aeq}=10Log\left(\frac{1}{\left(t_{2}-t_{1}\right)}\overset{t_{2}}{\underset{t_{1}}{\int }}\frac{p^{2}\left(t\right)}{p_{0}^{2}}\right)dt$$
where *p*(*t*) is the A-weighted instantaneous sound pressure, $p_{0}$ is the reference sound pressure equal to 20 × 10^−5^ N/m^2^, and *t*_1_, *t*_2_ are the time periods [50].

The hourly traffic volume is defined by the total number of cars that pass on a road section (e.g., intersection) in a specified interval of time. In addition, the total number of cars can be divided into car categories (vehicles classes). The average speed of the cars is set at 50 km/h.

The classification of vehicles and the calculation of average travel speeds was carried out by analyzing the video streams obtained from the video camera mounted on the drone and using an artificial convolutional neural network (CNN).

The road gradient is determined by the relationship:(11)$$Gradient=\frac{a-b}{L}$$
where the values of parameters *a*, *b*, and *L* are determined via topographic measurements.

The building density was calculated using the relationship:(12)$$Density=\frac{\sum_{i=1}^{n}\theta_{i}}{\theta_{t}}$$
where $\theta_{i}$ are the visibility angles for the facades of the buildings in the monitored area and $\theta_{t}$ is the maximum visibility angle (opening) of the video camera for the monitored area.

The tests were carried out in five residential areas of Bucharest, Romania, using a UAV on which was mounted an array of microphones, and the analysis was carried out at heights between 80 m and 110 m, for 15–20 min.

Figure 21 shows the result of testing the system using a sound source at a height of 100 m. Terms θ and ϕ represent the reception angles of the directional energy vectors for sound capture in relation to the horizontal and vertical movement of the drone on which the microphone array is mounted.

### 3.2. Collected Data Using the Artificial Neuronal Network

To carry out the data processing, a neural architecture with two analysis modules was used. The first module is the Supervised Convolutional Machine Learning (CNN), and the second module is an MFFN architecture for determining the traffic noise level. The structure is presented in Figure 22.

Considering the large amount of data associated with video information, a fusion system is proposed that uses deep neural networks to process the prediction output associated with multiple end-to-end systems, to combine all these outputs and create the set of predictions. Given the intrinsic ability of dense layers to detect patterns in the input data and to predict with high accuracy the analyzed samples, we theorize that, by using a set of dense layers, we can exploit the correlation between classifier biases, allowing different combinations of classifiers to strengthen or weaken the discriminative power of the pool of classifiers used in aggregation, based on the patterns learned by the dense network. In addition, because dense layers do not need special assumptions about the domain of the input data, we theorize that their use is also useful for creating a domain-independent aggregation system. The network architecture (Figure 23) is determined by constructing a search system designed to vary a set of dense network parameters as follows: (i) the optimal width by testing the following number of neurons per layer: {25, 50, 500, 1000, 2000, 5000}; (ii) network depth by changing the number of dense layers, testing the following values: {5, 10, 15, 20, 25}; (iii) adding or removing training subset normalization layers between dense layers.

First, there is no intrinsic spatial information associated with such an aggregate input (*y_i_),* therefore, some data decoration techniques must be developed to generate spatial correlations that can then be exploited by convolutional layers. As such, we create these correlations and relationships, through a process we call the input decorator. We theorize that by creating the decorated input vector for sample-level convolutional processing and applying convolutional filters to this new input, we can create a system where classifiers with similar discriminative powers are arranged in close spatial proximity, supporting or canceling group decisions based on their spatial relationships. Therefore, two problems need to be solved to introduce convolutional layers in aggregation networks: (i) finding a criterion for detecting similarity between classifiers and (ii) creating a similarity-based spatial organization system. Let $p_{j}=\left[y_{1j},\;y_{2j},y_{2j},\;\ldots ,\;y_{Mj}\right]\;$ be a prediction vector representing the outputs of classifier *j* for all *M* samples, where the similarity between two classifiers *m* and *n* can be calculated by *r_mn_* = *M* (*p_m_*, *p_n_*). Finally, by ordering the vector of similarity scores between a classifier m and all other classifiers, we can create a list of the most similar classifiers for each of the N classifiers. The second problem involves using the similarity values calculated in the previous step, and decorated using the given values of r, as follows:(13)$$\bar{d_{C_{i}}}=\left|\begin{array}{ccc} r_{4,1} & c_{1,1}\ldots & r_{1,N}\\ c_{4,1} & s_{1}\;\ldots & c_{2,N}\\ r_{3,1} & c_{3,1}\ldots & r_{2,N} \end{array}\right|$$

With the introduction of convolutional layers in the aggregation network, a method was created that can process the similarities between classifiers. However, convolutional networks have been proposed for image processing using the same filters for processing the entire image, and would, therefore, share, in the case of assembly systems, the same weights between different centroids. Although this is a step forward in classifier correlation processing, we theorize that this correlation between classifiers is different for each individual classifier, and, thus, weights should not be shared between centroids. With this assumption in mind, we propose to create a new type of DNN layer, which we call the “Cross-Space-Fusion”, or CSF layer. The implementation of the CSF layer is based on creating a new input decoration method as a new layer in the aggregation architecture.

Therefore, we propose a new input decoration method that creates an additional third dimension, which separately stores similar results of classifiers and similarity scores. Thus, the CSF layer processes these details in the third dimension of the matrix, processing the outputs of the classifiers and the similarity scores of the correspondents together, using the same M function for computing the similarity scores. We also need to quantify that regular convolutional filters may not be optimal for learning correlations, as they may be different from centroid to centroid. Thus, in the CSF layer, we design a larger number of parameters, and, although this leads to an additional depth of the neural network, the number of added parameters is still small, especially compared to the depth and width of the dense architecture. Considering these particularities, the following equation presents the operations performed by the new CSF layer:(14)$$\left[\begin{array}{ccc} \frac{\alpha_{1,i}s_{i}+\beta_{1,i}c_{1,i}r_{1,i}}{2} & \frac{\alpha_{2,i}s_{i}+\beta_{2,i}c_{2,i}r_{2,i}}{2} & \frac{\alpha_{3,i}s_{i}+\beta_{3,i}c_{3,i}r_{3,i}}{2}\\ \frac{\alpha_{8,i}s_{i}+\beta_{8,i}c_{8,i}r_{8,i}}{2} & S_{i} & \frac{\alpha_{4,i}s_{i}+\beta_{4,i}c_{4,i}r_{4,i}}{2}\\ \frac{\alpha_{7,i}s_{i}+\beta_{7,i}c_{7,i}r_{7,i}}{2} & \frac{\alpha_{6,i}s_{i}+\beta_{6,i}c_{6,i}r_{6,i}}{2} & \frac{\alpha_{5,i}s_{i}+\beta_{5,i}c_{5,i}r_{5,i}}{2} \end{array}\right]$$
where $c_{i}$ represents the matrix of predicted outputs for the eight most similar classifiers for a classifier *i*, while $R_{i}$ represents the respective similarity scores, calculated using the function M. These two matrices create the third dimension of the decorated input. After the decoration stage, the input is fed into the CSF layer. For each group ($c_{i},r_{i}$) of centroids, the network needs to create and learn a set of weights that combine the prediction of the original classifier with the prediction results and clustered similarity scores in the centroids. Thus, the CSF layer contains a set of parameters *α* and *β* to be learned, where parameters α are used to control the prediction output of each classifier *i* and parameters *β* are used to control the prediction outputs and similarity scores for classifiers similar to *i*.

The output of the CSF layer is then passed through a subsampling layer by the averaging function, yielding a new set of input data of the same size as the original. Two implementations were tested, in which pairs of contrasts were considered for the most similar four classifiers, and for the most similar eight classifiers.

To establish the performance values, it is necessary to calculate the confusion matrix. The confusion matrix consists of real values in one dimension and predicted labels for the second dimension, and each class consists of a row and a column. The diagonal elements of the matrix represent the correctly classified results.

The values calculated from the confusion matrix represent Precision, Recall, and F1-score, which is the harmonic average of the accuracy and recall and the accuracy of the classification:

Precision: it is defined as the number of samples that contain the existence of the drone.

Recall: it is defined as the ratio between the expected number of samples to contain a drone and the number of samples that contain the drone.

F-measure: it is defined as the harmonic average between accuracy and recall.

F1-scores are calculated for each class, followed by the average of the scaled scores with weights. The weights are generated from the number of samples corresponding to each class.

The network is trained for 200 epochs with a batch size of 16. This experiment resulted in a classification accuracy of 91 percent, Recall is 0.96, and micro average F1-score is 0.91. The confusion matrix is shown in Table 1 and the classification report is shown in Table 2.

### 3.3. Assessment of Noise Pollution in the Study Area

The evaluation of the noise levels at the measurement points, five residential areas in Bucharest, demonstrates the exceeding of the maximum permissible noise level for commercial-residential areas legislated in Romania, which is 60 dB(A), because the samples obtained showed a sound level that exceeds 75 dB(A), shown in Figure 24. Table 3 describes the statistical variables.

Analyzing the results of the $L_{aeq}$ analysis obtained for five residential areas in Bucharest, Romania, presented in Figure 24, it turns out that the permitted noise level is exceeded compared to the legislated level of 60 dB.

Noise pollution has increased to an alarming degree in all areas of the city (residential, commercial, industrial, and quiet) due to the rapid increase in urbanization, industrialization, and other connectivity of transport systems in all areas of the city. Identifying noise hotspots where immediate remedial action is required has always been a challenge. In addition to providing noise propagation in the X and Z directions, 2D noise mapping is an essential method for identifying regions where noise levels may reach a dangerous level. Thus, the comparative examination of all residential areas within a city provides a clear picture of noise exposure. The measurement of noise on the ground is carried out with the help of the sound level meter. In the horizontal direction, the orientation and distance of buildings from the road or flight path, as well as the shape of buildings are the main influencing factors on the noise effect. Based on the analyses carried out from ground level and from a height of 30 m, by making 2D maps, they showed that the surrounding structures (high buildings, brick walls, borders with grills, trees, etc.) significantly influence noise propagation in all directions. That is why the next step was to make 3D noise maps from a height of 100 m (e.g., road traffic and aircraft noise propagate from different directions and produce different effects on the ground floor and top floor of buildings). In particular, the buildings with the highest noise level are not close to the road or railway, and the apartments with high noise are not located close to the ground or the top of the building. The composition of the noise and the existence of multiple 3D noise sources determine the presence or absence of high noise. The high noise level in the residential area may be due to the heavy traffic volume and the road surrounded by tall buildings. 3D maps can also be used by decision makers in the process of formulating noise control strategies or implementing corrective measures.

## 4. Discussion

The novelties presented by the proposed solution are:➢Carrying out the analysis of noise pollution from traffic with the array of microphones mounted on a UAV.➢Use of the best algorithms reported in the field (energy, neural networks) for analyzing the solution.➢Use of a combined neural network structure, a CNN module for detecting the number of vehicles in traffic, calculating the average travel speed, and identifying vehicles (cars, trucks, and motorcycles) and the 2nd MFFN (multilayer feed-forward) module neural network for prediction.➢Proposal to create a new type of CNN layer, named “Cross-Space-Fusion”, or CSF layer. The implementation of the CSF layer is based on creating a new input decoration method as a new layer in the aggregation architecture.➢Proposing a new input decoration method for the CSF network, which creates an additional third dimension that separately stores similar results of classifiers and similarity scores. Thus, the CSF layer processes these details in the third dimension of the matrix, processing the outputs of the classifiers and the similarity scores of the correspondents together.➢To determine the level of noise pollution, the specialized literature only mentions measuring the ground level, and the solution proposed in the article can determine the sound level from a height (100 m), but also separate the individualized sound levels of vehicles in traffic.➢In the capital city of Romania, decibel limits are exceeded in many locations, every day. Unfortunately, the action plan for reducing noise is slow, or not developing, while the residents of the capital are increasingly affected by noise pollution. The level of noise and vibrations generated by the movement of means of transport belonging to the local transport society also have an impact, among the causes of the phenomenon being worn carriageways or the operation of a carriageway whose constructive solution does not correspond to the urban characteristics of the area (e.g., narrow streets, with fronts of buildings located near the roadway). There is a need for the development of an urban noise monitoring plan and adequate solutions for decreasing noise impact. The greatest impact is felt primarily by children and elderly people.

We believe that a solution to making noise reduction measures more efficient, while optimizing costs and efforts, could be the development of combined strategies to reduce traffic noise and air pollution.

## 5. Conclusions

Noise monitoring and prediction can raise various complex problems determined by non-linearity, and collected data are very large. The present research has been carried out in two phases: (i) collecting relevant information from traffic with traditional instruments, at the ground level and at an altitude of 30 m above and analyzing the data to determine patterns in noise behavior, and (ii) designing and testing a UAV-borne array of microphones and collecting similar data to determine feasibility of the solution. The solution is based also on previous research experience of the authors [55]. The proposed solution presented here is intended to solve many of these problems due to the use of neural network modules for the analysis of large volumes of data for noise prediction. Moreover, neural networks can be implemented in FPGA and VLSI, being superior compared to statistical and deterministic models. As future improvements and developments, we would like to enhance the filtering methods to suppress non-informative noise coming from the UAV’s own rotors [56,57,58] and lateral side noise sources by modifying the pattern of sensitivity in the horizontal plane, according to altitude of the UAV. Another concern is to choose the correct UAV type to keep environmental impact in urban sensitive areas (hospitals, residential areas, etc.) as low as possible [59]. In addition, a second direction of improvement that we are envisaging is to increase the directivity of the microphone array, based on recent research [60,61,62,63,64,65,66], and introduce the possibility to remotely configure it, according to local geometrical and environmental conditions.

## Figures and Tables

**Figure 1 sensors-23-01912-f001:**
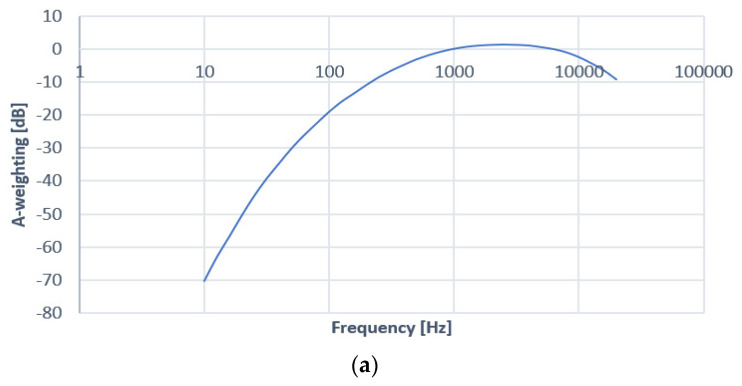
(**a**) The A-weighted curve for sound data compensation; (**b**) conversion from dB SPL to dBA using A-weighting adjustments.

**Figure 2 sensors-23-01912-f002:**
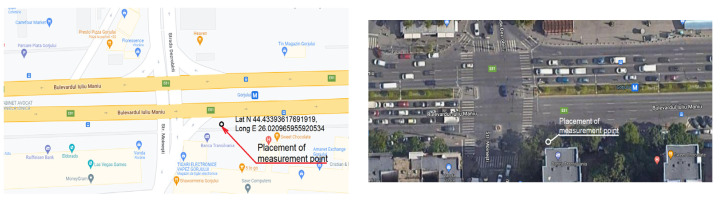
Placement of ground-level measurement point.

**Figure 3 sensors-23-01912-f003:**
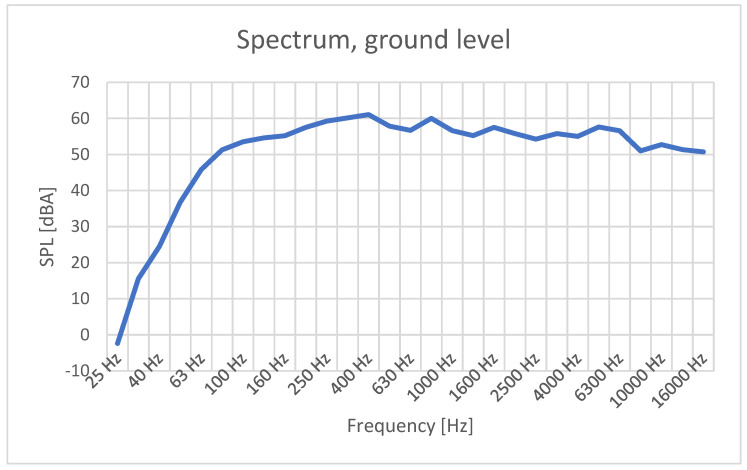
SPL versus frequency recorded in traffic-generated noise measurements, ground level.

**Figure 4 sensors-23-01912-f004:**
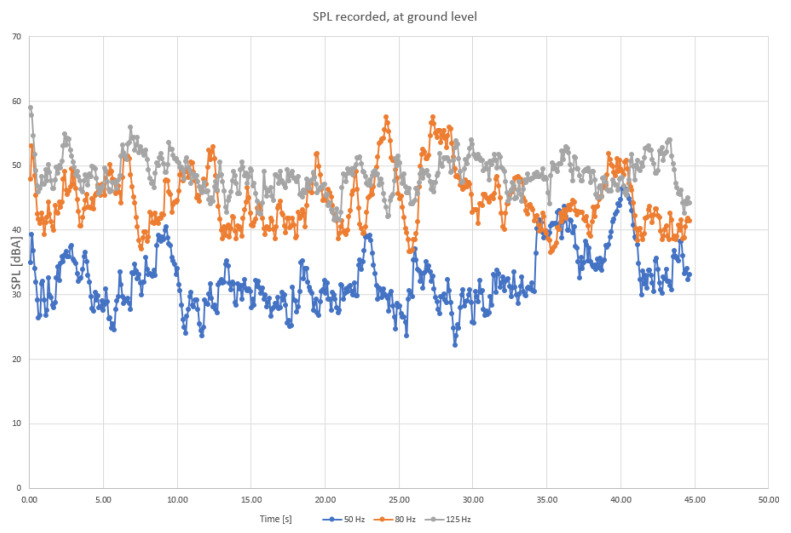
SPL variation recorded in lower frequency range (50 Hz, 80 Hz, and 125 Hz traffic noise), ground level: blue—50 Hz, orange—80 Hz, and grey—125 Hz.

**Figure 5 sensors-23-01912-f005:**
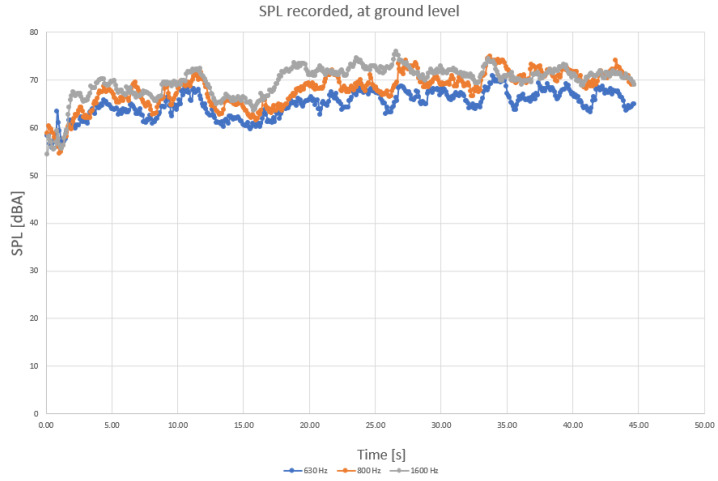
Recorded SPL at medium frequencies (630 Hz, 800 Hz, and 1.6 kHz traffic noise, in blue—630 Hz, orange—800 Hz, grey—1.6 kHz), ground level.

**Figure 6 sensors-23-01912-f006:**
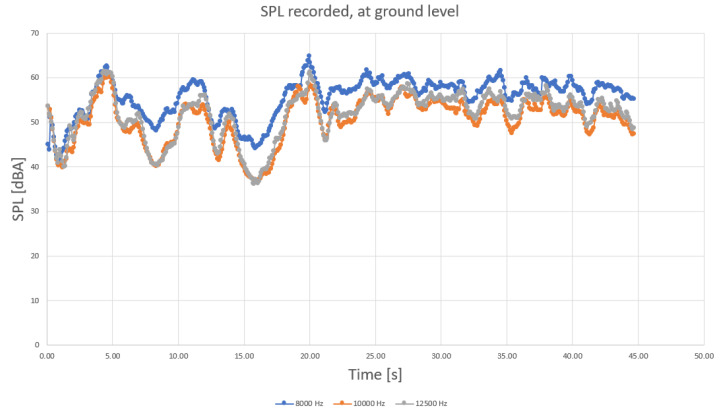
SPL recorded for higher frequency range, ground level (at 8 kHz, 10 kHz and 12.5 kHz, with blue—8 kHz, orange—10 kHz, and grey—12.5 kHz components).

**Figure 7 sensors-23-01912-f007:**
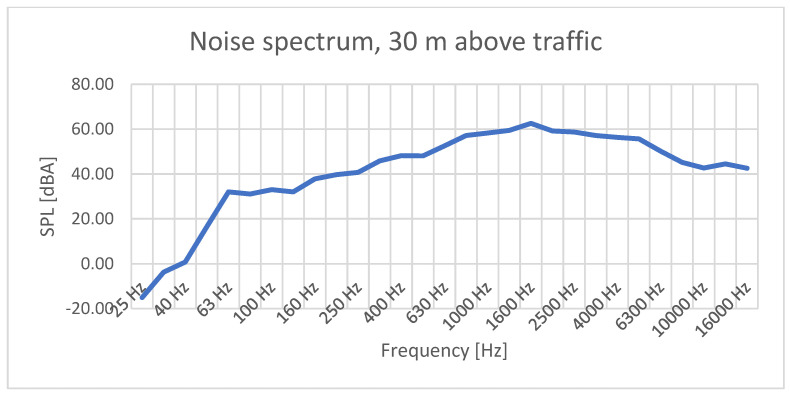
SPL versus frequency recorded in traffic-generated noise measurements, 30 m above ground.

**Figure 8 sensors-23-01912-f008:**
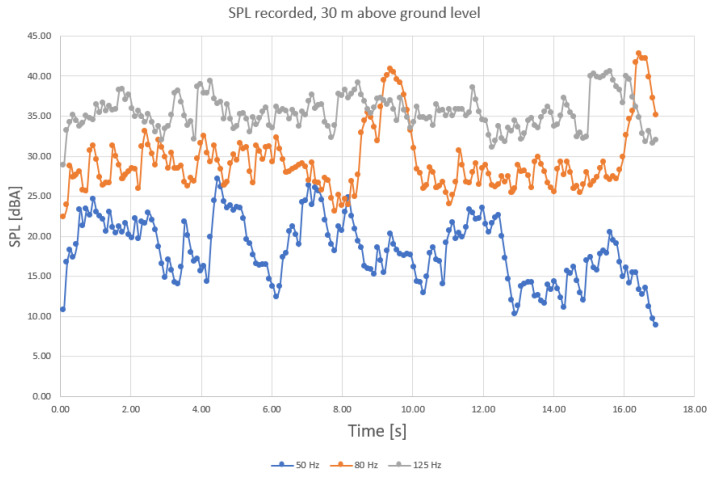
SPL recorded for higher frequency range, ground level (at 50 Hz, 80 Hz, and 125 Hz, with blue—50 Hz, orange—80 Hz, and grey—125 Hz components).

**Figure 9 sensors-23-01912-f009:**
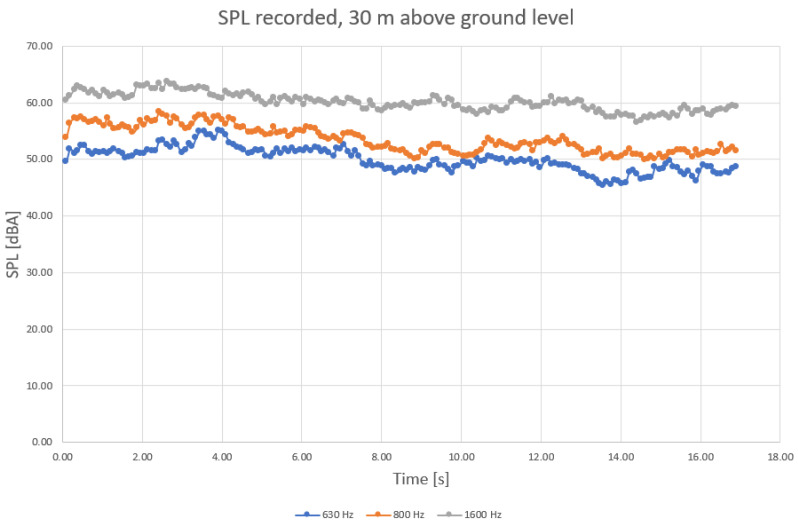
SPL recorded for medium frequency range, 30 m level (630 Hz, 800 Hz, and 1.6 kHz traffic noise, in blue—630 Hz, orange—800 Hz, grey—1.6 kHz).

**Figure 10 sensors-23-01912-f010:**
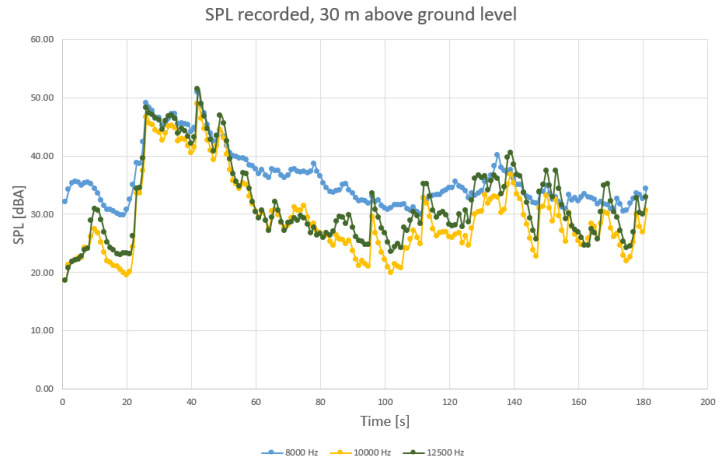
SPL recorded for medium frequency range, 30 m level (at 8 kHz, 10 kHz and 12.5 kHz, with blue—8 kHz, orange—10 kHz, and grey—12.5 kHz components).

**Figure 11 sensors-23-01912-f011:**
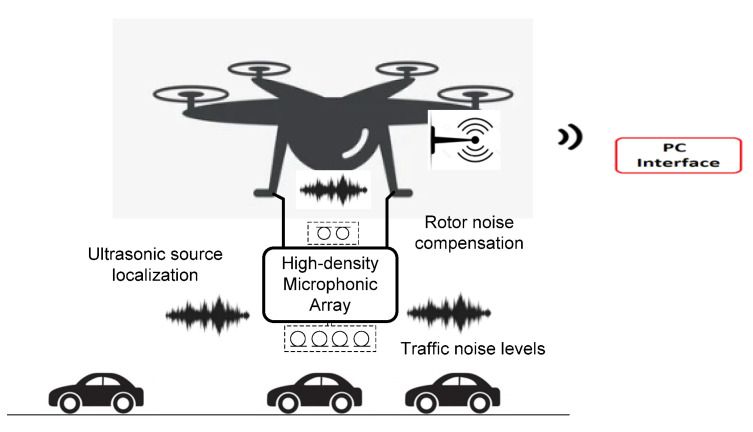
Architecture of the ultrasonic sound localization system using a drone.

**Figure 12 sensors-23-01912-f012:**
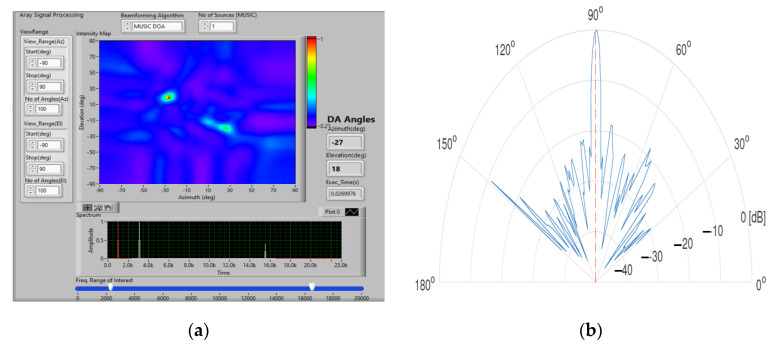
Simulation square microphone array and beamforming using acoustic frequency characteristic of UAVs (**a**) and directivity characteristic (**b**).

**Figure 13 sensors-23-01912-f013:**
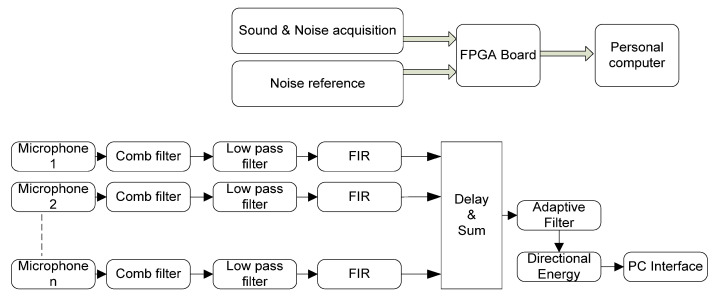
The acoustic processing system.

**Figure 14 sensors-23-01912-f014:**
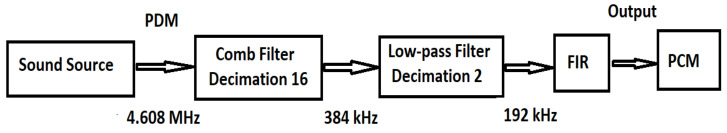
Convert from PDM to PCM.

**Figure 15 sensors-23-01912-f015:**
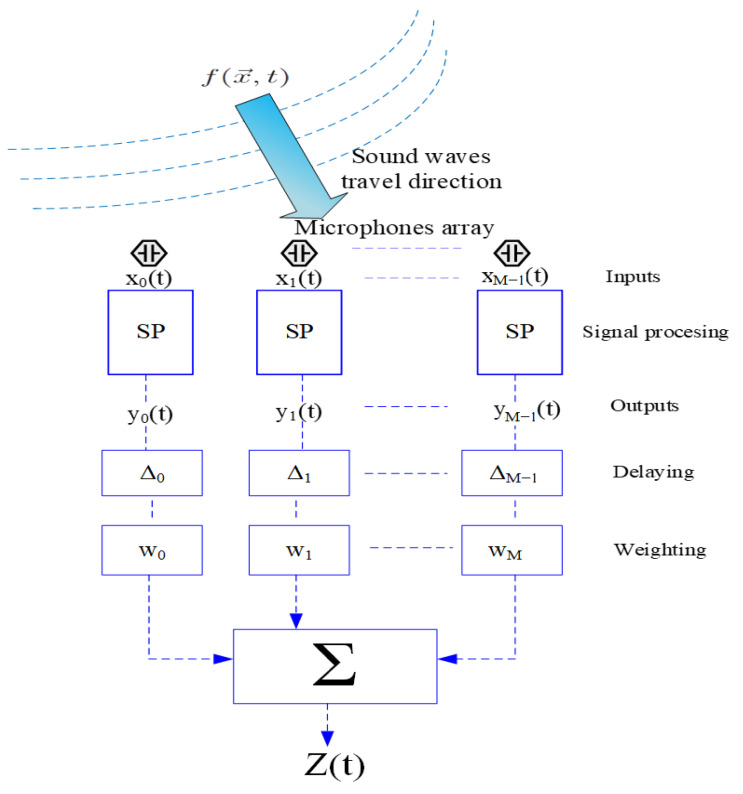
DAS beamforming algorithm working principle.

**Figure 16 sensors-23-01912-f016:**
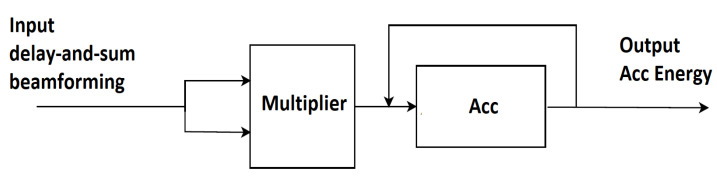
The method of realization and implementation for Equation (5).

**Figure 17 sensors-23-01912-f017:**
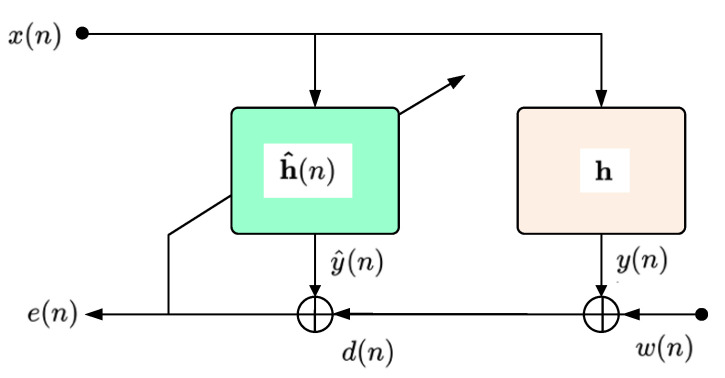
The system model.

**Figure 18 sensors-23-01912-f018:**
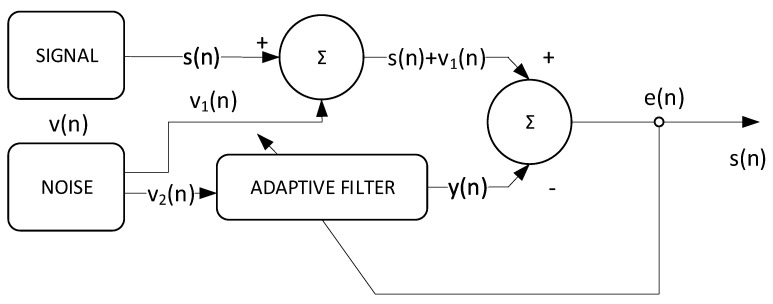
Adaptive noise cancellation theoretical configuration.

**Figure 19 sensors-23-01912-f019:**
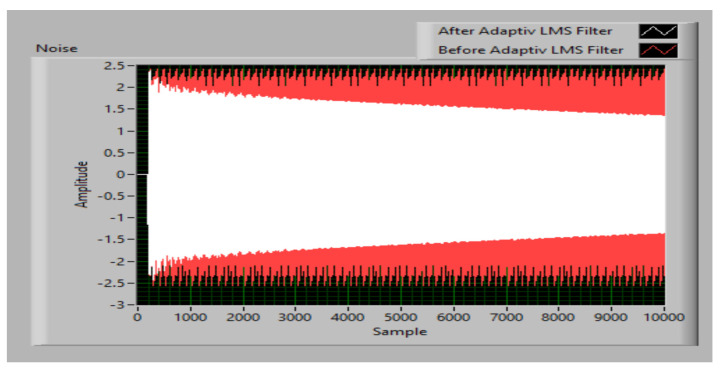
ANC simulation result with optimized LMS algorithm.

**Figure 20 sensors-23-01912-f020:**
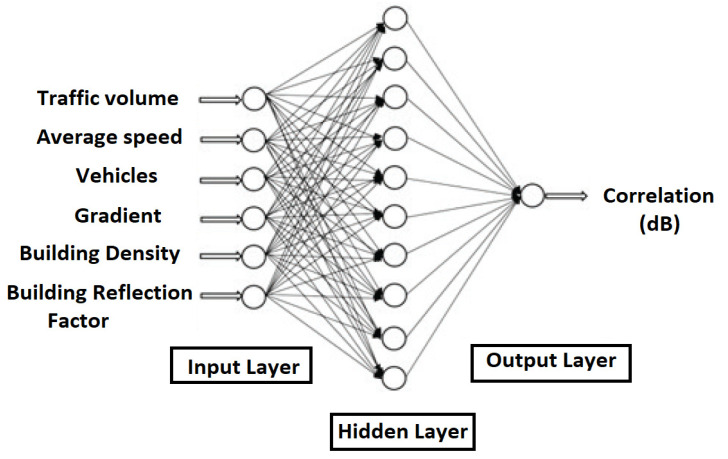
The proposed MFFC architecture for traffic noise analysis.

**Figure 21 sensors-23-01912-f021:**
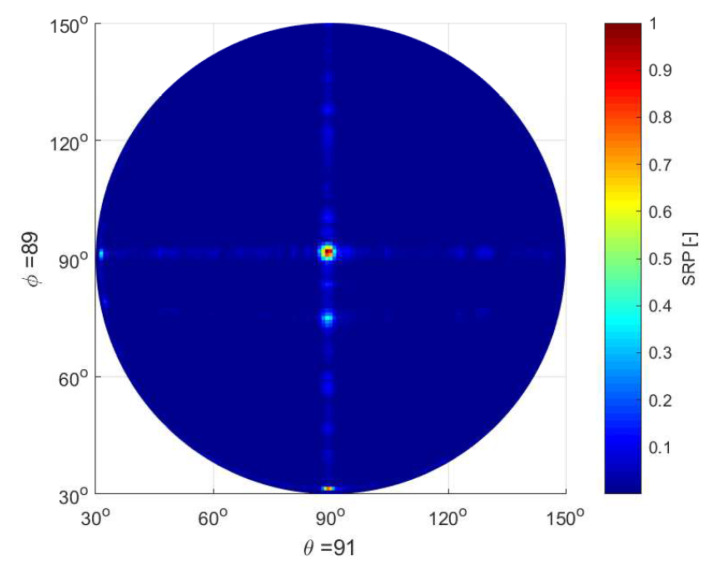
Result of testing the system for capturing sounds from a height of 100 m.

**Figure 22 sensors-23-01912-f022:**
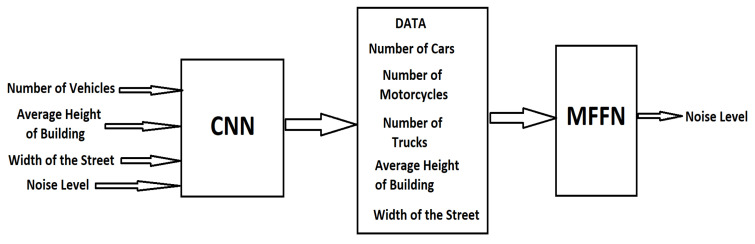
The neural architecture proposed.

**Figure 23 sensors-23-01912-f023:**
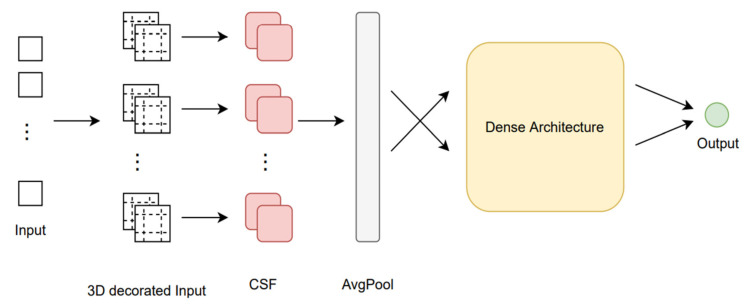
Cross-Space-Fusion aggregation network diagram.

**Figure 24 sensors-23-01912-f024:**
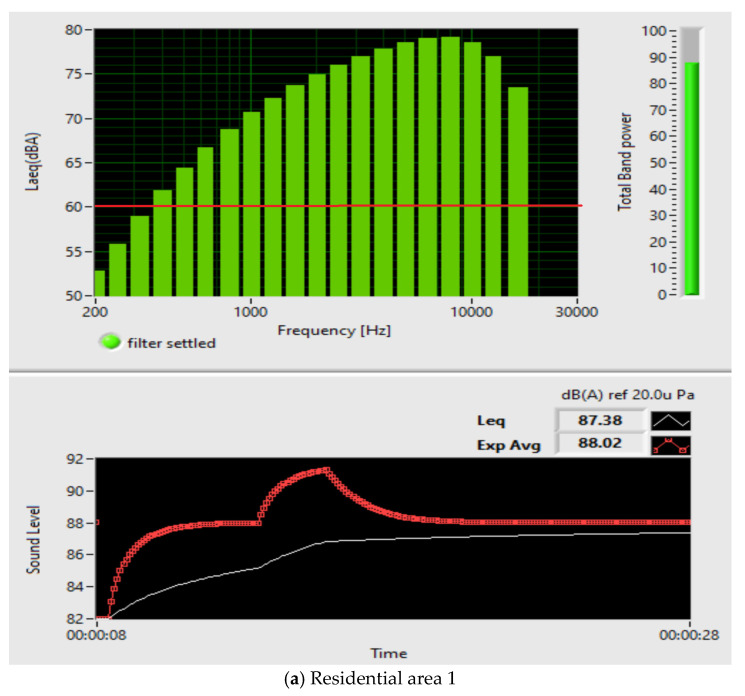
Noise pollution levels determined for five residential areas in Bucharest, Romania.

**Table 1 sensors-23-01912-t001:** Confusion matrix.

		Predict Label		
		Background Noise	Single Car	Two Cars
True Label	Background Noise	1.00	0.00	0.00
	Single Car	0.00	0.94	0.06
	Two Cars	0.00	0.23	0.77

**Table 2 sensors-23-01912-t002:** Classification report.

Classes	Precision	Recall	F1-Score
Background Noise	1	1	1
Single Car	0.80	0.94	0.87
Two Cars	0.93	0.77	0.84
Avg/total	0.91	0.90	0.90

**Table 3 sensors-23-01912-t003:** Descriptive statistics of variables.

	Traffic Volume	Average Speed	No. of Motor	No. of Trucks	No. of Cars	G	D	*L_aeq_*
Mean	4556.78	59.367	5.4122	4.1414	6.5632	2.992	0.75	74.669
Standard error	274.78	2.587	0.270	0.145	0.83	0.435	0.045	0.426
	2106.3	19.97	1.945	1.037	1.647	2.047	0.269	3.952
Min.	1302	24.437	0.759	0.367	1.448	0.04	0	74.57
Max.	9759.5	94.978	8.826	4.523	9.565	9	0.99	87.38

## Data Availability

The data used were obtained in the laboratory of Intelligent Transport Systems within the Faculty of Transport, Polytechnic University of Bucharest. The data obtained are not public.

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
