# Peer review of "Urban Traffic Noise Analysis Using UAV-Based Array of Microphones"

_sensors, 2023, doi:10.3390/s23041912_

Round 1
Reviewer 1 Report
Dear Authors,
The article entitled “Urban traffic noise analysis using UAV-based array of microphones” presents original research about the monitoring and mapping of 3D traffic generated urban noise emissions using a simple UAV-based, and low-cost solution. The overall paper quality is good, and I think it can be interesting to the journal readers but needs to be improved before publication. Please allow me to offer a couple of suggestions in more detail.
The paper has a well-written introduction, which sets out the argument and summarizes the research related to the topic. The gaps in the current understanding are clearly underlined. The authors should reference recent literature.
The research carried oud involves sufficient use of control experiments, statistical analysis, and samplings. The tables and figures aid in comprehending the research.
The tendencies that were seen are analyzed, and the importance of the findings are discussed, in the Results and Discussion sections.
To improve the content's natural flow and make it easier for potential readers to grasp, the conclusion may be enhanced. Please expand on the authors' theoretical and practical implications in the Conclusion section.
Please describe the research's future direction.
The bibliographic basis is large and well-done, but it might use a few more references from the last five years to make it more current.
Editing mistakes should be corrected in the paper.
I hope you will find these recommendations to be useful.
I wish you success in all your activities!
Author Response
Thank you for your comments and helpful suggestions. Please find attached our responses.

Reviewer 2 Report
The introduction is fragmented and missing consistency. I recommend reformulation. Also, the author start by discussing urban pollution then move to noise pollution without linking them together.
In materials, author discuss SPL and pa as unit, then provide discuss on the weighting by dB without discussing the relationship between them., please reconsider this part.
Citations are missing in section 2, please reconsider this aspect.
Line 237, health effects, this comes late in the paper, it requires further discussion and validation through literature and in the introduction.
All different subfigures in Figures 3, 4, and 5, could plotted in the same graph for better comparison.
At first,the authors discussed measurements at ground level and at 30m height. Then the author discuss the implementation and measurements utilizing UAV without comparing or relating to their previous measurements. This should be reconsidered as it might proove important to compare results between these approaches.
Author Response
Thank you for your comments and useful suggestions. Please find attached our responses.

Round 2
Reviewer 2 Report
The be version of this document is acceptable